# The *Cis*-Effect Explained Using Next-Generation QTAIM

**DOI:** 10.3390/molecules27186099

**Published:** 2022-09-18

**Authors:** Yuting Peng, Wenjing Yu, Xinxin Feng, Tianlv Xu, Herbert Früchtl, Tanja van Mourik, Steven R. Kirk, Samantha Jenkins

**Affiliations:** 1Key Laboratory of Chemical Biology and Traditional Chinese Medicine Research and Key Laboratory of Resource National and Local Joint Engineering Laboratory for New Petro-chemical Materials and Fine Utilization of Resources, College of Chemistry and Chemical Engineering, Hunan Normal University, Changsha 410081, China; 2EaStCHEM School of Chemistry, University of St Andrews, North Haugh, St Andrews, Fife KY16 9ST, Scotland, UK

**Keywords:** QTAIM, NG-QTAIM, cis-effect, dihaloethene, dihalodiazene

## Abstract

We used next-generation QTAIM (NG-QTAIM) to explain the *cis*-effect for two families of molecules: C_2_X_2_ (X = H, F, Cl) and N_2_X_2_ (X = H, F, Cl). We explained why the *cis*-effect is the exception rather than the rule. This was undertaken by tracking the motion of the bond critical point (*BCP*) of the stress tensor trajectories T_σ_(*s*) used to sample the U_σ_-space cis- and trans-characteristics. The T_σ_(s) were constructed by subjecting the C1-C2 BCP and N1-N2 BCP to torsions ± θ and summing all possible T_σ_(s) from the bonding environment. During this process, care was taken to fully account for multi-reference effects. We associated bond-bending and bond-twisting components of the T_σ_(s) with *cis-* and *trans-*characteristics, respectively, based on the relative ease of motion of the electronic charge density *ρ*(**r_b_**). Qualitative agreement is found with existing experimental data and predictions are made where experimental data is not available.

## 1. Introduction

An extensive body of literature exists on the subject of the *cis*-effect over the last 40 years, on the basis of both theory [1,2,3,4,5,6] and experiment [6,7,8,9,10,11,12,13,14,15,16,17]. The *cis*-effect is a phenomenon in which the *cis*-isomer is more stable than [2,4,7,8,9] or, in molecules with double bonds, almost as energetically stable as the corresponding *trans*-isomer. Computational results of the relative energy differences of the *cis*- and *trans*-isomers are compared with existing experimental data [7,8,9,10,11,12,13,14,15,16] that in some cases suffer from large uncertainties in the measurements [7,15,16,17]. In this study, we will not use energy-based approaches to attempt to understand the mechanism of the *cis*-effect; instead, we will use an electronic charge-density-based analysis.

Earlier, some of the current authors provided a *scalar* physics-inspired coupling mechanism explaining the *cis*-effect in terms of electronic and nuclear degrees of freedom for three families of molecules, including halogen-substituted ethene and diazenes [18]. We undertook a *static* investigation of the properties of the central X = X or N = N bond paths and found that those of the *cis*-isomers were more bent: the difference between the length of the bond path and the internuclear separation was up to 1.5% larger in the *cis*-isomers than in the corresponding *trans*-isomers. In our earlier contribution, we therefore concluded that the physical origin of the *cis*-effect was associated with greater bond-path bending. This earlier work, however, only provided correlations of the bond-path bending with the energy and did not explain why the *cis*-effect is the exception rather than the rule. In this work, the physical basis of the *cis*-effect will be provided in terms of the least and most preferred directions of electronic charge density motion.

Recently, some of the current authors used an electronic charge-density-based analysis to investigate steric effects within the formulation of next-generation Quantum Theory of Atoms in Molecules (NG-QTAIM) [19]. We found that the presence of chiral contributions suggested that steric effects, rather than hyperconjugation, explained the staggered geometry of ethane [20]. This recent work on steric effects relates to the current investigation on cis-effects, since in both cases we subject the central C = C or N = N bond to a torsion to probe either steric or *cis*-effects. Low/high values of the NG-QTAIM interpretation of chirality (C_σ_) were associated with low/high steric effects due to the absence/presence of an asymmetry. The chirality C_σ_ [21] was earlier used to redefine a related quantity for cumulenes, the bond-twist T_σ_ [22].

In this investigation, we will use NG-QTAIM to explain why the *cis*-effect was previously found, in our scalar investigation, to be associated with bond bending [18]. This will be undertaken by subjecting the axial bonds, C1-C2 and N1-N2, to a torsion θ to sample the directional response of the electronic charge density *ρ*(r**_b_**) at the bond cross-section. This will provide a better understanding of the greater (topological) stability of the *cis*-isomer over the *trans*-isomer in these halogen-containing species; see Figure 1.

## 2. Theoretical Background and Computational Details

### 2.1. Theoretical Background 

The QTAIM and next-generation QTAIM (NG-QTAIM) [23,24,25,26,27,28,29] background is provided in the Appendix A, including the procedure to generate the stress tensor trajectories T_σ_(s).

We use Bader’s formulation of the quantum stress tensor σ(r) [30] to characterize the forces on the electron density distribution in open systems that is defined by:(1)$$\sigma \left(r\right)=-\frac{1}{4}{\left[\left(\frac{\partial^{2}}{\partial r_{i}\partial r_{j}^{\prime }}+\frac{\partial^{2}}{\partial r_{i}^{\prime }\partial r_{j}}-\frac{\partial^{2}}{\partial r_{i}\partial r_{j}}-\frac{\partial^{2}}{\partial r_{i}^{\prime }\partial r_{j}^{\prime }}\right)\cdot \gamma \left(r,r^{\prime }\right)\right]}_{r=r^{\prime }}$$
where *γ*(**r**,**r**′) is the one-body density matrix,
(2)$$\gamma \left(r,r^{\prime }\right)=N\int \Psi (r,r_{2},\ldots ,r_{N})\Psi^{*}\left(r^{\prime },r_{2},\ldots ,r_{N}\right){\mathrm{dr}}_{2}\cdots {\mathrm{dr}}_{N}$$

The stress tensor is then any quantity σ(r) that can satisfy equation (2): any divergence-free tensor can be added [30,31,32]. Bader’s formulation of the stress tensor σ(r), equation (1), is a standard option in the AIMAll QTAIM package [33]. Earlier Bader’s formulation of σ(r) demonstrated superior performance compared with the Hessian of *ρ*(r) for distinguishing the *S*_a_- and *R*_a_-geometric stereoisomers of lactic acid [34] and therefore will be used in this investigation.

In this investigation, we include the entire bonding environment, including all contributions to the U_σ_-space *cis-* and *trans-*characteristics, by considering the C1-C2 *BCP* T_σ_(s) of the asymmetric, i.e., ‘reference’ carbon atom (C1) or the N1-N2 BCP T_σ_(s) of the nitrogen atom (N1); see Figure 1 and the Computational Details section. The C1-C2 *BCP* T_σ_(s) and N1-N2 BCP T_σ_(s) are created by subjecting these BCP bond paths to a set of torsions θ; see the Computational Details section.

The bond-twist T_σ_ is the difference in the maximum projections, the dot product of the stress tensor e_1σ_ eigenvector and the BCP displacement dr, of the T_σ_(s) values between the counter-clockwise (CCW) and clockwise (CW) torsion θ.
T_σ_ = [(e_1__σ_∙dr)_max_]_CCW_ − [(e_1__σ_∙dr)_max_]_CW_
(3)

Equation (3) for the bond-twist T_σ_ quantifies the bond torsion *BCP*-induced *bond twist* for the CCW vs. CW direction, where the largest magnitude stress tensor eigenvalue (λ_1**σ**_) is associated with e_1σ_; see Figure 1 and Figure 2. The eigenvector e_1σ_ corresponds to the direction along which electrons at the *BCP* are subject to the most compressive forces. Therefore, e_1σ_ corresponds to the direction along which the *BCP* electrons will be displaced most readily when the *BCP* is subjected to a torsion [35]. Higher values of the bond twist T_σ_ correspond to greater asymmetry, and therefore to a dominance of the *trans*- compared with the *cis*-isomer in U_σ_ space. This reflects the structural symmetry, with respect to the positioning of the halogen substituents, of the *trans*- rather than the *cis*-isomer.

Conversely, the eigenvector e_2σ_ corresponds to the direction along which the electrons at the *BCP* are subject to the least compressive forces. Therefore, **e_2σ_** corresponds to the direction along which the *BCP* electrons will be least readily displaced when the *BCP* undergoes a torsion distortion. The *bond-flexing* F_σ_ associated with **e_2σ_** is defined as:F_σ_ = [(e_2__σ_∙dr)_max_]_CCW_ − [(e_2__σ_∙dr)_max_]_CW_
(4)

The bond-flexing F_σ_ is calculated from the torsion *BCP bond flexing* defined by equation (4); see Figure 1 and Figure 2. Equation (4) provides a U_σ_-space measure of the ‘flexing-strain’ or bond bending that a bond path is under in the *cis-* or *trans*-isomer configurations. This is consistent with greater ‘flexing-strain’ or bond bending that previously correlated with a greater presence of the *cis*-effect [18]. Higher values of F_σ_ correspond to the dominance of the *cis*- compared with the *trans*-isomer in U_σ_ space, because bond bending reflects the symmetry of the *cis*-isomer rather than the *trans*-isomer, with respect to the positioning of the halogen substituents.

The bond-axiality A_σ_ is part of the U_σ_-space distortion set ∑{T_σ_,F_σ_,A_σ_}, which provides a measure of the chiral asymmetry. It is defined as:A_σ_ = [(e_3__σ_∙dr)_max_]_CCW_ − [(e_3__σ_∙dr)_max_]_CW_
(5)

Equation (5) quantifies the direction of *axial* displacement of the bond critical point (*BCP*) in response to the bond torsion (CCW vs. CW), i.e., the sliding of the *BCP* along the bond path. We will, however, not use A_σ_ as it does not comprise the bond cross-section, but provide it in the Appendix A. Instead, we will use the so-called U_σ_-space *bond cross-section* set ∑{T_σ_,F_σ_} developed for *cis*- and *trans*-isomers.

The (+/−) sign of the bond-twist T_σ_ and bond-flexing F_σ_ determines the S_σ_ (T_σ_ > 0, F_σ_ > 0) or R_σ_ (T_σ_ < 0, F_σ_ < 0) character; see Table 1 and Table 2.

The bond cross-section set ∑{T_σ_,F_σ_} is related to the cross-section of a *BCP* bond path that is quantified by the λ_1σ_ and λ_2σ_ eigenvalues associated with the e_1σ_ and e_2σ_ eigenvectors, respectively. Note, the e_1σ_ and e_2σ_ eigenvectors are the directions along which the *BCP* electrons are displaced most readily and least readily, respectively, when the *BCP* is subject to a torsional distortion. The *trans*-isomer is dominant in U_σ_ space if the magnitude of the bond-twist T_σ_ value is larger for the *trans*- than for the *cis*-isomer. Conversely, dominance of the *cis*-isomer is determined by the presence of a larger magnitude bond-flexing F_σ_ value for the *cis*-isomer compared with the *trans*-isomer; see Table 1 and Table 2.

### 2.2. Computational Details

The electronic wavefunction for molecular structures incorporating a chemically conventional double bond is usually well-represented by a single-reference wavefunction in the ‘eclipsed’ configurations 0° (*cis*) or 180° (*trans*). It is also well-known that as the dihedral angle across the double bond deviates from the ‘eclipsed’ configurations, the nature of the wavefunction changes, becoming fully multi-reference in nature at the twisted 90° ‘staggered’ configuration. The multi-reference character is determined for ethene using the frequently used T1 measure [36], where values of T1 > 0.02 indicate that a single-reference description is inadequate. For this reason, in all of this work, for both the C_2_X_2_ (X = H, F, Cl) and N_2_X_2_ (X = H, F, Cl) molecules, we use a multi-reference CAS-SCF(2,2) method [37,38], using Slater determinants for the active space, implemented in Gaussian G09.E01 [39] with symmetry disabled, an ‘ultrafine’ integration grid and convergence criteria of ‘VeryTight’ geometry convergence and an SCF convergence criterion of 10^−12^. The cc-pVTZ triple-zeta basis set was used during geometry optimization and the dihedral coordinate scan constrained geometry optimization process. The magnitude of the dihedral angle scan steps was 1°. Additionally, the second atom used in each sequence defining the dihedral scan angle, C1 and N1, respectively, for the ethene and diazene derivatives, was constrained to be fixed at the origin of the Cartesian spatial coordinates. All initial ‘eclipsed’ (*cis*- and *trans*-) optimized molecular geometries were generated (and checked to be energy minima with no imaginary vibrational frequencies) with these settings; see Appendix A for the optimized structures and tabulated experimental data. The final single-point wavefunctions and densities for each structure produced during the dihedral scans were calculated, as recommended for accurate NG-QTAIM properties [40], using a quadruple-zeta basis set (cc-pVQZ).

The direction of torsion is determined to be CCW (0.0° ≤ θ ≤ +90.0°) or CW (−90.0° ≤ θ ≤ 0.0°) from an increase or a decrease in the dihedral angle, respectively. An exception is made for N_2_Cl_2_ where the respective angular limits used were −80° and +80°. These latter limits are chosen due to the well-known destabilizing interactions between the nitrogen lone pairs and the relatively weak N-Cl bonds [41,42], which cause dissociation of the molecule into N_2_ and Cl_2_ when a larger twist is applied: we observed and confirmed this dissociation for dihedral twists > 80°.

The stress tensor trajectories T_σ_(s) for all four of the possible ordered sets of four atoms that define the dihedral angle for each of the *cis*-isomers (X3-C1-C2-X6, X3-C1-C2-H5, H4-C1-C2-H5, H4-C1-C2-X6) and *trans*-isomers (X4-C1-C2-X6, X4-C1-C2-H5, H3-C1-C2-H5, H3-C1-C2-X6) are calculated; see the left panel of Figure 1 for the dihedral atom numbering. There is a single dihedral X3-N1-N2-X4 for each of the *cis-* and *trans*-isomers of the diazenes N_2_X_2_ (X = H, F, Cl); see the right panel of Figure 1.

QTAIM and stress tensor analysis were then performed on each single-point wavefunction obtained in the previous step with the AIMAll [33] and QuantVec [43] suite. In addition, all molecular graphs were confirmed to be free of non-nuclear attractor critical points.

## 3. Results and Discussions

The scalar distance measures geometric bond length (GBL) and bond path length (BPL) used in this investigation on ethene, doubly substituted ethene and diazene are insufficient to quantify the presence of the *cis*-effect and are provided in the Appendix A. The variation in the (scalar) relative energy ∆E for ethene, doubly substituted ethene and diazene molecules do not provide any insights either into the *cis*-effect for these molecules and are provided in the Appendix A. The intermediate and the complete C1-C2 *BCP* U_σ_-space distortion sets are provided in the Appendix A, respectively.

The sum of the bond-cross-section sets ∑{T_σ_,F_σ_} of the C1-C2 *BCP* T_σ_(s) was calculated for all four possible isomers of the formally achiral molecules ethene and doubly substituted ethane, C_2_X_2_ (X = H, F, Cl). The results for the molecular graph of pure ethene are provided as a control to enable a better understanding of the effect of the halogen atom substitutions; see Table 1 and Figure 1. The corresponding results for the formally achiral diazenes N_2_X_2_ (X = H, F, Cl) comprising a single isomer N1-N2 *BCP* T_σ_(s) are presented in Table 2. The magnitude of the values of the bond cross-section set ∑{T_σ_,F_σ_} increases with atomic weight, as is demonstrated for F_2_ and Cl_2_ substitution of ethene; see Table 1. This relationship between the magnitude of the ∑{T_σ_,F_σ_} values and halogen substituent also occurred in a recent investigation of singly and doubly halogen-substituted ethane [44]. This dependency of {T_σ_,F_σ_} on the atomic weight of the substituent, however, does not occur for the diazenes.

The magnitude of the bond-twist T_σ_ is significantly smaller for the *cis*- compared with the *trans*-isomer for C_2_X_2_ (X = F, Cl) and slightly smaller for N_2_X_2_ (X = Cl). The magnitude of the bond-flexing F_σ_ is significantly larger for the *cis*- compared with the *trans*-isomer for C_2_X_2_ (X = F, Cl) and N_2_X_2_ (X = F, Cl).

These results for the bond cross-section set ∑{T_σ_,F_σ_} are consistent with the presence of the *cis*-effect and therefore indicate the occurrence of the *cis*-effect in U_σ_ space for C_2_X_2_ (X = F, Cl) and N_2_X_2_ (X = F, Cl). The very large component of the bond-twist T_σ_ for N_2_X_2_ (X = H) indicates a complete lack of the *cis*-effect and a dominance of the *trans*-isomer in U_σ_ space for this molecule.

All of the investigated molecular graphs comprised a significant degree of chiral character as indicated by the magnitudes of the bond-twist T_σ_ and bond-flexing F_σ_, particularly for C_2_X_2_ (X = F, Cl).

## 4. Conclusions

In this investigation, NG-QTAIM was used to determine the presence or absence of the *cis*-effect for the C_2_X_2_ (X = H, F, Cl) and N_2_X_2_ (X = H, F, Cl) molecules. Qualitative agreement with experimental data for differences in the energies of the *cis-* and *trans*-isomers was found.

The molecules of this investigation are formally achiral according to the Cahn–Ingold–Prelog (CIP) priority rules [45], but all comprise at least a degree of chiral character in U_σ_ space, on the basis of the magnitude of the T_σ_ values, with C_2_X_2_ (X = Cl) displaying a very significant degree of chirality. This finding reflects the conventional understanding that steric effects are among the reasons for the differences between the relative energetic stabilities of *cis-* and *trans*-isomers, consistent with our previous association of chiral character in Uσ space for the steric effects for ethane [20].

We found that both C_2_X_2_ (X = F, Cl) and N_2_X_2_ (X = F, Cl) display the *cis-*effect. This includes the prediction of a cis-effect in N_2_X_2_ (X = Cl), for which no experimental data on the *cis*-isomer and *trans*-isomer energy difference are available. The *cis*-effect is determined on the basis of the much larger values of the bond-flexing Fσ for the *cis*- compared with the *trans*-isomer.

We provided a physical explanation as to why the *cis*-effect is the exception rather than the rule, by defining a dominant bond-flexing F_σ_ component of the bond cross-section set {T_σ_,F_σ_} as characterizing the *cis*-effect. This is on the basis that it is more difficult to bend (F_σ_) than to twist (T_σ_) the C1-C2 *BCP* bond path and N1-N2 *BCP* bond path. This difference in the difficulty of performing bond-bending (F_σ_) and bond-twisting (T_σ_) distortions is explained by their construction, using the least preferred e_2σ_ and most preferred e_1σ_ eigenvectors, respectively, that determine the relative ease of motion of the electronic charge density *ρ*(r_b_).

Suggestions for future work include the exploration of the newly discovered NG-QTAIM bond cross-section set {T_σ_,F_σ_} for *cis-* and *trans*-isomers, which could be undertaken by manipulating the *cis-* and *trans*-isomer character in U_σ_ space with laser irradiation. We make this suggestion since NG-QTAIM chirality has already been found to be reversed by the application of an electric field [46]. Reversing the *cis-* and *trans*-isomer character in U_σ_ space is possible with laser irradiation that is fast enough to avoid disrupting atomic positions. Such a reversal in U_σ_ space could result in the *cis-* and *trans*-geometric isomers comprising *trans-* and *cis*-isomer assignments in U_σ_ space, respectively.

## Data Availability

Not applicable.

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
