# Peer review of "The Cis-Effect Explained Using Next-Generation QTAIM"

_molecules, 2022, doi:10.3390/molecules27186099_

Round 1

Reviewer 1 Report

In the manuscript, QTAIM and the stress tensor analysis of molecular distortion are applied in cis-effect on the substituted ethenes.

Except for the more advanced method applied, the manuscript brings not much new data or insights compared to the paper (ref.18) previously published by the same research group.

The manuscript is not easy to follow. An effort to better explain the physical meaning of BCP stress tensor trajectories and their connection with the isomer energy would be desirable for wider audience.

Page 7 lines 244-250, the statements are somewhat speculative.

Supplementary materials are appropriate and extensive, and in part already published along with the papers by the same research group.

The authors might consider using QTAIM and the stress tensor analysis rather than NG-QTAIM.

Minor points:

Check equations 1-3 numbering and referencing.

Complete Reference 18 and 31.

Page 2, lines 80-83: please rephrase and clarify the sentence.

Figs. 1 and 2: use a), b) and c) notation rather than „sub-figures“.

Shorten extensive use of „Supplementary Materials” description. i.e. instead of “in the Supplementary Materials S5 and Supplementary Materials S6” use “in the Supplementary Materials S5 and S6”.

Please avoid referencing to Supplementary Materials in Conclusions.

Author Response

In the manuscript, QTAIM and the stress tensor analysis of molecular distortion are applied in cis-effect on the substituted ethenes.

Except for the more advanced method applied, the manuscript brings not much new data or insights compared to the paper (ref.18) previously published by the same research group.

We explained why the cis-effect is the exception rather than the rule, this is entirely new and a significant contribution to the understanding of this important chemical effect. Previously (ref 18) we only correlated the cis-effect as occurring when there is a higher degree of cis- compared with trans-isomer bond-path bending in a static investigation. This earlier attempt did not provide any details into the nature of the cis-effect including why it is the exception. In this work we do more than provide simple correlations in that we explain why the cis-effect is relatively rare in terms of it requiring a greater distortion of the least preferred (along the eigenvector e) than the most preferred direction (along the eigenvector e) of charge density accumulation. The e eigenvector corresponds to the direction along which electrons at the BCP are subject to the most compressive forces. Therefore, the e eigenvector corresponds to the direction along which the BCP electrons will be displaced most readily when the BCP is subjected to a torsion. The converse is true for the e eigenvector.

Therefore, the following clarification is added to the introduction to explain limitations of the earlier work (ref 18):

This earlier work however, only provided correlations of the bond-path bending with the energy and did not explain why the cis-effect is the exception rather than the rule.

The manuscript is not easy to follow. An effort to better explain the physical meaning of BCP stress tensor trajectories and their connection with the isomer energy would be desirable for wider audience.

We do not correlate the stress tensor trajectories with the isomer energy. Instead, in this contribution we explain the cis-effect in terms of mechanics by creating a measure of the dominance of preference of the motion of the electronic charge density by the bond-twist Tσ(s) associated with e (the most preferred direction) and the bond-flexing Fσ associated with e (the least preferred direction). The cis-effect occurs where the magnitude of the bond-flexing Fσ is greater than that of the bond-twist Tσ(s).

In the conclusions we already wrote:

“We provided a physical explanation as to why the cis-effect is the exception rather than the rule, by defining a dominant bond-flexing Fσ component of the bond cross-section set {Tσ,Fσ} as characterizing the cis-effect. This is on the basis that it is more difficult to bend (Fσ) than to twist (Tσ) the C1-C2 BCP bond-path and N1-N2 BCP bond-path. This difference in difficulty of performing bond-bending (Fσ) and bond-twisting (Tσ) distortions is explained by their construction, using the least preferred e and most preferred e eigenvectors, respectively, that determine the relative ease of motion of the electronic charge density ρ(rb).”

We have, however, added some clarification in the introduction:

In this work the physical basis of the cis-effect will be provided in terms of the least and most preferred directions of electronic charge density motion.

Page 7 lines 244-250, the statements are somewhat speculative.

This section contains suggestions for future work. To make this clearer we replace the original text:

Further exploration of the newly discovered NG-QTAIM bond cross-section set {Tσ,Fσ} for cis- and trans-isomers could be undertaken by manipulating the cis- and trans-isomer character in Uσ-space with laser irradiation. Reversing the cis- and trans-isomer character in Uσ-space with laser irradiation fast enough to avoid disrupting atomic positions is in principle possible. Such a reversal in Uσ-space would result in the cis- and trans-geometric isomers predominately comprising trans- and cis-isomer assignments in Uσ-space, respectively.

With the clarified text:

Suggestions for future work include the exploration of the newly discovered NG-QTAIM bond cross-section set {Tσ,Fσ} for cis- and trans-isomers, which could be undertaken by manipulating the cis- and trans-isomer character in Uσ-space with laser irradiation. We make this suggestion since NG-QTAIM chirality has already been found to be reversed by the application of an electric field[46]. Reversing the cis- and trans-isomer character in Uσ-space with laser irradiation fast enough to avoid disrupting atomic positions is in principle possible. Such a reversal in Uσ-space would result in the cis- and trans-geometric isomers predominately comprising trans- and cis-isomer assignments in Uσ-space, respectively.

Supplementary materials are appropriate and extensive, and in part already published along with the papers by the same research group.

We thank the reviewer for taking notice of the Supplementary Materials.

The authors might consider using QTAIM and the stress tensor analysis rather than NG-QTAIM.

NG-QTAIM is defined to be a directional analysis constructed from eigenvectors that is necessary for the current analysis. Our earlier work (ref 18) that considered only static cis- and trans-isomers used conventional (scalar) QTAIM and the stress tensor could not recover the bond-twist or bond-flexing. The stress tensor analysis that we do is part of NG-QTAIM in the form of the stress tensor trajectories Tσ(s).

Minor points:

Check equations 1-3 numbering and referencing.

The equation numbering has been corrected.

Complete Reference 18 and 31.

  1. Jenkins, S.; Kirk, S.R.; Rong, C.; Yin, D. The Cis-Effect Using the Topology of the Electronic Charge Density. Molecular Physics 2012, 111, 1–13, doi:10.1080/00268976.2012.745631.

Reference 31 is now renumberd as reference 33:

  1. Keith, T.A. AIMAll (19.10.12), TK Gristmill Software, Overland Park KS, USA (Http://Aim.Tkgristmill.Com) 2019.

This has been undertaken:

Page 2, lines 80-83: please rephrase and clarify the sentence.

The original text of lines 80-83:

We use Bader’s formulation of the stress tensor[30], a standard option in the AIMAll QTAIM package[33], which is used in this investigation because of the superior performance of the stress tensor compared with the Hessian of ρ(r) for distinguishing the Sa- and Ra-geometric stereoisomers of lactic acid[34].

Has been replaced by the corrected text:

Bader’s formulation of the stress tensor σ(r) is a standard option in the AIMAll QTAIM package[33] and is used in this investigation because of the superior performance compared with the Hessian of ρ(r) for distinguishing the Sa- and Ra-geometric stereoisomers of lactic acid[34].

With the added reference:

  1. Li, Z.; Nie, X.; Xu, T.; Li, S.; Yang, Y.; Früchtl, H.; van Mourik, T.; Kirk, S.R.; Paterson, M.J.; Shigeta, Y.; et al. Control of Chirality, Bond Flexing and Anharmonicity in an Electric Field. International Journal of Quantum Chemistry 2021, 121, e26793, doi:10.1002/qua.26793.

Figs. 1 and 2: use a), b) and c) notation rather than „sub-figures“.

This has been corrected.

Shorten extensive use of „Supplementary Materials” description. i.e. instead of “in the Supplementary Materials S5 and Supplementary Materials S6” use “in the Supplementary Materials S5 and S6”.

This has been corrected.

Please avoid referencing to Supplementary Materials in Conclusions.

This has been corrected.

Reviewer 2 Report

1. Authors should check and fully revise the article, especially since the similarity to citation 20 (11% excluding biography) seems high.

2. The resolutions of Figures 1 and 2 should be improved.

3. Introduction: Enough

4. Theoretical Background and Computational Details

# The authors used citations 41-43 for the information “… the stress tensor without violating this definition 79 [41–43]”, but I couldnt see citations 42 and 43 in the References section. They used 41 citations totally.

# They have to revise the citation of “AIMAll [31]”, citation 31 seems incomplete.

# They have to give the citation for "CAS-SCF(2,2) method"

5. Conclusion section:

# The statement "see the Supplementary Materials S2" in the first passage should be deleted.

# The authors used some discussions in this section. The important discussions have to be moved to the "Results and Discussion" section, and this section should include only the main key results that are obtained from all computations. The second passage (includes citations 41 and 20) should be moved to the relevant section.

6. Supplementary Material: This supporting file have to be revised according to the journal guidelines.

# This supporting file has to be revised according to the journal guidelines.

# The resolution of Figure S6 should be improved.

# Table S2 has to be re-prepared: The authors prepared this Table directly using the output data (cut & copy).

# They used some citations in this section but did not use the References section in the Supplementary Material. They have to revise accordingly. 

Round 2

Reviewer 2 Report

Dear Editor,

Authors revised text as suggested.

I recommend it to be ACCEPT.